# Growth Inhibition by Amino Acids in *Saccharomyces cerevisiae*

**DOI:** 10.3390/microorganisms9010007

**Published:** 2020-12-22

**Authors:** Stephanie J. Ruiz, Joury S. van ’t Klooster, Frans Bianchi, Bert Poolman

**Affiliations:** Department of Biochemistry, Groningen Biomolecular Sciences and Biotechnology Institute, University of Groningen, Nijenborgh 4, 9747 AG Groningen, The Netherlands; s.j.ruiz@outlook.com (S.J.R.); j.s.van.t.klooster@rug.nl (J.S.v.t.K.); f.bianchi@rug.nl (F.B.)

**Keywords:** amino acid transport, amino acid toxicity, growth inhibition, *Saccharomyces cerevisiae*

## Abstract

Amino acids are essential metabolites but can also be toxic when present at high levels intracellularly. Substrate-induced downregulation of amino acid transporters in *Saccharomyces cerevisiae* is thought to be a mechanism to avoid this toxicity. It has been shown that unregulated uptake by the general amino acid permease Gap1 causes cells to become sensitive to amino acids. Here, we show that overexpression of eight other amino acid transporters (Agp1, Bap2, Can1, Dip5, Gnp1, Lyp1, Put4, or Tat2) also induces a growth defect when specific single amino acids are present at concentrations of 0.5–5 mM. We can now state that all proteinogenic amino acids, as well as the important metabolite ornithine, are growth inhibitory to *S. cerevisiae* when transported into the cell at high enough levels. Measurements of initial transport rates and cytosolic pH show that toxicity is due to amino acid accumulation and not to the influx of co-transported protons. The amino acid sensitivity phenotype is a useful tool that reports on the in vivo activity of transporters and has allowed us to identify new transporter-specific substrates.

## 1. Introduction

As well as being the building blocks for proteins, amino acids provide raw materials for energy generation, nitrogen metabolism, and the biosynthesis of structural, signaling, or defensive compounds [1]. Although they are essential metabolites, it has been known for over half a century that the addition of excess amino acids to both prokaryotic and eukaryotic cell cultures can cause growth inhibition and/or cell death [2,3,4,5,6,7,8,9]. The dependence of toxicity on transport activity indicates that the effect is exerted intracellularly [10,11]. In humans, several inherited metabolic diseases are caused by elevated levels of amino acids and/or closely related metabolites, with perhaps the most well-known being phenylketonuria [12].

Some amino acids are known to cause toxicity via specific mechanisms. Valine and phenylalanine inhibit the growth of bacterial cells by repressing enzymes involved in synthesis of isoleucine and tyrosine, respectively [13,14]. Histidine toxicity in yeast cells is caused by a reduction in copper availability in vivo [11]. There is also evidence for a more general mechanism involving the target of rapamycin complex 1 (TORC1), a master controller of cell metabolism that is conserved among eukaryotes and responds to various nutrient stimuli, including amino acids [15,16,17]. Deregulation of TORC1 signaling is linked to many human diseases [18], and it has been recently shown that in some cancer cell lines, this deregulation is caused by abnormal amino acid transport [19,20,21]. Phenylalanine sensitivity in a mammalian cell line, and rescue by valine, has been shown to involve mTORC1 [22]. All amino acids except glutamine are growth inhibitory to the plant *Nicotiana silvestris,* and glutamine itself can rescue some (but not all) of the inhibition caused by other amino acids [23,24,25]. Glutamine is now known to be an important activator of TORC1 [26,27,28].

Wild-type yeast cells are able to synthesize all amino acids de novo but can also scavenge them from the environment using a host of broad- and narrow-range transporters located in the plasma membrane [29,30]. *Saccharomyces cerevisiae* amino acid permeases have been studied extensively not only in terms of their transport activity but also their regulation. Many are transcriptionally up- or downregulated by three interconnected pathways that respond to the availability of amino acids and other nitrogen sources: nitrogen catabolite repression (NCR), general amino acid control (GAAC), and SPS-signaling [29,31,32,33]. Some transporters are also post-translationally regulated in response to the external concentration of their substrates e.g., Can1 (Arg), Dip5 (Glu), Gap1 (various), Lyp1 (Lys), Mup1 (Met), and Tat2 (Trp) [34,35,36,37,38]. In this pathway, substrate binding triggers transporter ubiquitination and subsequent removal of the proteins from the plasma membrane via endocytosis [39]. This causes a decrease in transport activity and thus limits the accumulation of certain substrates. Similar regulation has been observed for Ptr2, which imports di- and tri-peptides that can be broken down into free amino acids inside the cell [40]. It has been suggested that this is a mechanism to avoid amino acid-induced toxicity [40,41].

Risinger et al. [41] showed that *S. cerevisiae* cells expressing Gap1^K9R,K16R^, a ubiquitination- and endocytosis-deficient mutant that is constitutively localized to the plasma membrane, experience severe growth defects when individual amino acids are added to the medium at high concentrations (3 mM). The same effect, with varying levels of severity, was triggered by the metabolite citrulline as well as all proteinogenic amino acids except alanine. We decided to investigate the amino acid sensitivity of strains constitutively overexpressing eight different narrow- and broad-range amino acid transporters: Agp1, Bap2, Can1, Dip5, Gnp1, Lyp1, Put4, or Tat2 (Table 1).

## 2. Materials and Methods

### 2.1. Strains and Growth Conditions

*Escherichia coli* strain MC1061 (*E. coli* Genetic Stock Center) was used for cloning and plasmid storage. Experiments were performed with *S. cerevisiae* strains BY4709 (*MAT*α *ura3Δ0*) and BY4741 (*MATa hisΔ1 leu2Δ0 met15Δ0 ura3Δ0*) [61]. BY4709 was used for growth experiments and transport assays, while BY4741 was used for measurements of the internal pH.

All experiments were done in YNBD, a minimal medium containing 6.9 g/L Yeast Nitrogen Base without amino acids (Formedium^TM^, Norfolk, UK), 20 g/L D-glucose (Sigma-Aldrich, St. Louis, MO, USA), and 100 mM of potassium phosphate (pH 6.0). Single amino acids or a standard synthetic complete (SC) mixture (Kaiser Drop-out minus uracil, Formedium^TM^) were added for growth assays. For pHluorin experiments a low fluorescence version of YNBD was made using Yeast Nitrogen Base without amino acids and without folic acid and riboflavin (Formedium^TM^), and methionine and histidine were added at 76 mg/L each (509 and 490 μM, respectively). For growth assays using Arg and Lys at concentration ≤500 μM, the YNBD did not contain potassium phosphate buffer, but the pH was set to 6.0 using HCl/NaOH before sterilization by filtration. All media contain ammonium sulfate (5 g/L) as nitrogen source.

Agar was added at 20 g/L for solid cultures. Liquid cultures (5–15 mL) were inoculated from single colonies on agar plates and incubated in 50 mL CELLreactor^TM^ filter top tubes (Greiner Bio-One, Kremsmünster, Austria) at 30 °C with shaking (~200 rpm).

### 2.2. Plasmids

The plasmids used in this study are listed in Table 2. Vectors for the constitutive expression of transporters are based on pFB022 and pFB023 [62]. These are pRS426 derivatives (*URA3*, 2μ) that allow for expression of fluorescently-tagged Lyp1, either full length or truncated, from the constitutive ADH1 promoter [63]. The C-terminal tag contains a TEV protease recognition site, followed by the fluorescent protein YPet [64] and an eight-residue His epitope (for the full sequence, see the Appendix A). Plasmids pSR045–051 are identical to pFB022 except for replacement of *LYP1* with other transporter genes as indicated in Table 1. They were constructed by in vivo homologous recombination using crude PCR products. The plasmid backbone was amplified from pFB022 [62], using primers binding immediately upstream and downstream of the *LYP1* coding region. The reaction was then treated with *Dpn*I (as per the manufacturer’s instructions) to remove the template DNA. Each target gene was PCR amplified from *S. cerevisiae* BY4742 chromosomal DNA, using forward and reverse primers that added approximately 25 bp of sequence homologous to the plasmid backbone (Table 3). The backbone and insert were simultaneously transformed into BY4709 and positive transformants screened by growth on uracil dropout media, colony PCR and fluorescence microscopy. Fusion genes were confirmed by sequencing of the entire open reading frame. A single base pair mutation (T to C) was observed at the end of the ADH1 promoter, but this did not appear to affect expression.

Ratiometric pHluorin [67] was expressed from pYES2-*P_ACT1_*-pHluorin [66]. For these experiments a truncated Lyp1 with no fluorescent tag was used [67]. First, the C-terminal YPet tag was removed from pFB023 using uracil excision-based cloning [68]. Primer pairs 4158/3631, 3630/5159 and 5087/5089 (Table 3) and the PfuX7 polymerase [69] were used to amplify pFB023 in three fragments excluding the YPet coding region. The crude PCR products were treated with USER^TM^ enzyme (New England Biolabs, Ipswich, MA, USA) as per the manufacturer’s instructions and transformed into *E. coli* MC1061 for in vivo assembly. After the resulting plasmid (pSR053) was confirmed by sequencing of the entire coding region, the area containing *LYP1_(62-590)_* and the flanking promoter and terminator sequences were sub cloned into pRSII425 using *Sac*I/*Kpn*I digestion to yield plasmid pSR057. The expressed protein (from N- to C-terminus) includes a starting Met, residues 62-590 of Lyp1, a short three-residue glycine linker, and an eight-residue His epitope.

### 2.3. Microscopy

Live cell imaging was performed on a LSM 710 commercial laser scanning confocal microscope (Zeiss, Oberkochen, Germany) equipped with a C-Apochromat 40×/1.2 NA objective. Cells were immobilized between a glass slide and coverslip. Images were obtained with the focal plane positioned at the mid-section of the cells. For fluorescence images, samples were excited with a blue argon ion laser (488 nm) and emission collected at 493–797 nm.

### 2.4. Growth Assays

Growth assays were performed using 120 µL liquid cultures in CELLSTAR^®^ 96-well flat-bottom microplates (Greiner Bio-One). Each plate was covered with a Breathe-Easy^®^ sealing membrane (Sigma-Aldrich), as well as the provided lid, and incubated in a 30 °C room at 400 rpm on an Excella™ E1 benchtop open-air shaker (Eppendorf, Hamburg, Germany). OD_600_ measurements were made in a PowerWave 340 spectrophotometer (BioTek, Winooski, VT, USA) without the microplate lid but with the (optically clear) membrane.

To prepare inocula for microplate experiments, strains were cultured in YNBD for approximately 24 h, with one round of dilution, to an OD_600_ of 0.3–0.7. Each culture was diluted in fresh media to an OD_600_ of 0.1 and 60 µL aliquots were added to microplate wells containing 60 µL of media with or without either single amino acids or the SC mixture. The final concentrations are given in Table 4 and the relevant figure legends. All measurements were done in biological triplicate (three independent inoculations made on different days from different pre-cultures).

Raw OD_600_ values were background corrected by subtracting the average value for all blank (media only) wells from the same plate (*n* = 12–24). This observed value (OD_obs_) was then corrected for non-linearity at higher cell densities using the polynomial equation: OD_cor_ = 0.319 × OD_obs_^3^ + 0.089 × OD_obs_^2^ + 0.959 × OD_obs_ (Appendix A); this correction method was previously discussed in Warringer and Blomberg [70]. OD_cor_ values were then normalized within each strain and replicate. Three independent experiments were performed but, due to technical error, some results had to be discarded. For this reason, *n* = 2 for Bap2 1 mM Gly/His/Ile/Leu/Lys/Met/Orn/Phe, and also for all Tat2 1 mM and 0.5 mM conditions.

### 2.5. Transport Assays

In vivo transport assays were performed as previously described [62]. The radioactive substrates used were L-(^14^C(U))-phenylalanine, L-(^14^C(U))-lysine and L-(methyl-^3^H)-methionine (PerkinElmer, USA) and L-(^14^C(U))-alanine (Amersham Biosciences, Little Chalfont, Buckinghamshire, UK). Transport was assayed at the following concentrations: 50 μM lysine with and without 100 mM of histidine or ornithine, 500 μM alanine or methionine, and 2.5 mM phenylalanine.

### 2.6. Measurement of Cytosolic pH

Strains were cultured in liquid media to an OD_600_ of 0.3–0.7, diluted to an OD_600_ of 0.2 in pre-warmed media, and then immediately transferred to the spectrophotometer sample holder (also pre-warmed). After 15 min, 10 µL of either distilled water or 100 mM lysine was added (final concentration 500 µM) and the measurement continued for another 3 h.

Fluorescence measurements were made using a JASCO FP-8300 fluorescence spectrophotometer with the following settings: sensitivity, high; response, 0.1 s; excitation/emission bandwidths, 5 nm; emission, 508 nm; excitation, 355–495 nm in 5 nm steps. All samples were 2 mL in 4.5 mL plastic cuvettes (catalogue number 1961, Kartell Labware, Noviglio, Italy) with a magnetic bar for stirring. The temperature of the sample holder was maintained at 20 °C for calibration measurements and 30 °C for experiments with growing cells.

To prepare calibration samples, strains were cultured as above. Cells were then washed twice and resuspended in 5 volumes of ice-cold PBS (10 mM Na_2_HPO_4_, 137 mM NaCl, 2.7 mM KCl, 1.8 mM KH_2_PO_4_, pH 7.35). Digitonin was added to a final concentration of 0.02% *w*/*v* (from a 2% *w/v* stock in PBS) and the samples incubated for 1 h at 30 °C with mixing. Cells were then washed twice with 1 mL of ice-cold PBS, resuspended to 0.25 mg/mL with the same (1 µL added per mg wet weight of cell pellet) and kept on ice. 10 µL of cells was added to 2 mL of room-temperature buffer (either 100 mM potassium phosphate or PBS) and a fluorescence measurement made once per minute for ten minutes.

We found that background subtraction was very important, especially for long-term measurements with growing cells. Strains carrying pHluorin/pRSII425 or pHluorin/Lyp1_(62-590)_ were compared to strains carrying pRSII426/pRSII425 and pRSII426/Lyp1_(62-590),_ respectively. To generate the calibration curve (Appendix A), the median emission intensities at 395 and 475 nm excitation (I_395_ and I_475_) were background subtracted, and the ratio (R_395/475_) plotted against the pH. For measurements with growing cells, the I_395_ and I_475_ values from each strain and timepoint were background subtracted using individual values from the corresponding strain and timepoint.

## 3. Results and Discussion

### 3.1. All Standard Proteinogenic Amino Acids, as Well as Ornithine and Citrulline, Inhibit Growth of S. Cerevisiae

Eight different *S. cerevisiae* amino acid transporters were overexpressed by introducing their genes on multicopy plasmids under control of the constitutive ADH1 promoter (Figure 1). A C-terminal tag containing the fluorescent protein YPet (https://www.fpbase.org/protein/ypet/) was added to allow visualization of the transporters in the cell. Confocal fluorescence microscopy confirmed that all eight transporters localize primarily to the cell periphery (Figure 1A), indicating that our tagged proteins are delivered to the plasma membrane. Some fluorescent signal is seen in internal membranes and the vacuole, which is not uncommon for plasma membrane transporters, given that their normal life cycle involves being trafficked between these different locations [39,71]. We observed substantial variation in protein expression between cells from the same culture, with a significant proportion showing little or no fluorescence (Appendix A). We believe that this reflects a sub-population of cells that have lost the expression vector. For the high copy pRS plasmids it is known that plasmid-free cells make up 20–30% of the population, even in selective media [72]. Expression of particular proteins can increase this population significantly, with reported values of up to 50% [73]. Using flow cytometry, we found that the fraction of cells in the fluorescent population (30–65%) and the median fluorescence signal from individual cells in this population (1.5–5.6 arbitrary units) was reasonably stable between independent cultures of each strain (Figure 1B,C). In minimal media without amino acids, growth of the overexpression strains over 24 h was up to 40% lower than that of BY4709 containing the empty plasmid pRSII426 (Figure 1D). This is likely due to both the plasmid instability discussed earlier (different populations of plasmid-free cells translates to a difference in the starting cell density of each culture), as well as general effects caused by overexpression [74,75]. These variations within and between strains do not affect the conclusions of this paper as we only make comparisons of the same strains grown in different conditions.

We compared the growth of each strain in minimal medium with or without the addition of either single amino acids or a synthetic complete (SC) mixture (Figure 2). It should be noted that BY4709 contains all the endogenous *S. cerevisiae* amino acid transporters and is in the yeast S288C background. We therefore expect that at the start of the experiment all strains, which were precultured in minimal medium containing ammonium as the sole nitrogen source, would contain significant levels of active Gap1 [76,77]. For the control strain, the SC mixture increased growth and only cysteine or histidine at 5 mM was inhibitory. For all overexpression strains except Put4, at least two different individual amino acids (other than cysteine or histidine) inhibited growth by more than 50%. We observed no citrulline-mediated growth inhibition, consistent with the fact that Gap1 is the only citrulline transporter [78]. Risinger et al. [41] observed no alanine-mediated toxicity and only a 20% reduction in cell growth caused by phenylalanine (all other amino acids caused a ≥65% reduction in growth). Here, we show that 5 mM of phenylalanine reduces the growth of Agp1 and Gnp1 strains by 70%, while 5 mM of alanine reduces the growth of Agp1 strains by 30%. Ornithine, a basic amino acid and important metabolite, inhibited the growth of strains overexpressing Can1 and Lyp1.

We were surprised that proline or valine did not cause higher levels of growth inhibition. Both of these amino acids caused a 90% reduction in growth when supplied at 3 mM to an *S. cerevisiae* strain expressing a endocytosis-resistant Gap1 mutant [41]. Agp1, Gnp1, and Put4 are all proline transporters, with Put4 estimated to have an approximately 500-fold lower *K*_m_ for proline than Gap1 [45,56]. Valine is known to be a substrate for both Agp1 and Bap2, with transport in sub-mM concentrations occurring at rates equal or higher to that of other substrates for which we did observe growth inhibition (e.g., Ile, Leu) [43,47,60]. Sensitivity to other amino acids and increases in whole cell uptake (Figure 2 and Figure 3A) indicate that Agp1, Bap2, and Gnp1 are all present in these strains. The largest effect caused by proline or valine, however, was an approximately 10% reduction in the growth of Agp1 strains. It is possible that this difference is due to variations in substrate-regulated trafficking. Although the use of a constitutive promoter minimizes transcriptional regulation, the levels of transporter at the plasma membrane and thus the rate of substrate transport and accumulation is subject to post-translational regulation via intracellular trafficking [34,35,36,38]. A series of elegant experiments using Can1 and Gap1 mutants showed that this process requires ligand binding but not transport, supporting the hypothesis that substrate binding induces conformational changes that promote endocytosis [38]. The same study demonstrated that different substrates are more or less effective at triggering endocytosis of the same transporter, and it follows that the same amino acid could be more or less effective at triggering endocytosis of different transporters.

### 3.2. Amino Acid Sensitivity Reports on Transporter Activity and Substrate Specificity

We expected that the pattern of amino acid sensitivity for each strain would match the known substrate specificity (Table 1) of the overexpressed transporter and, for the most part, this was true (Figure 2). As discussed in the previous section, some known substrates did not cause substantial growth inhibition. Even more surprising was that many strains were sensitive to amino acids not predicted to be transport substrates. While it is possible that these results are affected by altered expression levels of endogenous systems, we tentatively conclude that these transporters are able to transport a much broader range of substrates than previously described, but with a high Michaelis constant (*K*_m_ > 1 mM). This can be rationalized by structural and mutational studies, which indicate that the residues interacting with the α-amino and α-carboxyl groups of the substrate are conserved [49,51,79]. The reason why this broader specificity has not been seen before could be simply because previous studies have tested a limited range of substrates and/or concentrations. The screening study published by Regenberg et al. [60], for example, only assayed transport at 100 or 250 µM of substrate. These lower-affinity transport activities would be important to consider in a laboratory setting where synthetic media often contains amino acids at (sub-)mM concentrations.

Some of these novel substrate specificities (the transport of Phe by Gnp1 and of His and Orn by Lyp1) were further investigated by monitoring the transport of radioactive substrates into whole cells (Figure 3). Again, it should be noted that some endogenous transporters, including Gap1, are expected to be present under these conditions but rapidly endocytosed in response to amino acid addition. For this reason, we only measured initial rates of transport. Overexpression of Gnp1 indeed increased the uptake of phenylalanine into whole cells (Figure 3A), supporting its identification as a phenylalanine transporter. Lysine uptake was 4-fold faster in cells overexpressing Lyp1 (Figure 3B,D), indicating that the YPet-tagged Lyp1 is active. When His or Orn was present at 100 mM (2000-fold excess) Lys transport by Lyp1 cells was reduced by 56% and 43%, respectively. At the same concentrations, Lys transport by the control strain was reduced to background levels (Figure 3B). These results, in conjunction with the growth inhibition, are consistent with Lyp1 transporting both His and Orn. They also suggest that Lyp1 has a lower affinity for these substrates than Can1, which has a reported inhibition constant (*K*_i_) of 3 mM for both His and Orn [80], or Gap1. It has previously been reported that Lyp1 does not transport Orn or His, but this could be explained by a lower affinity and the fact that competition for Lys transport by Lyp1 was assayed with only a ten-fold excess in comparison to the 1000-fold excess used in the competition for Arg transport by Can1 [50,80]. Subsequent studies, using Lyp1 overexpression strains, tested for His transport using concentrations of 50–100 μM, which is likely to be far below the *K*_m_ [54,60].

In further experiments using lower concentrations, Lyp1 strains were inhibited (≥50% reduction in growth) by 16 µM lysine and Can1 strains by 16 µM of arginine or 125 µM of lysine (Figure 4). This is in the range of the measured *K*_m_ values (Table 1) [51,62]. The growth advantage seen for Lyp1 overexpression strains in the presence of external arginine is due to import of this amino acid, which can be used as a carbon or nitrogen source by other endogenous transporters. The same effect is observed for the control strain, although it is only apparent during the exponential growth phase (Appendix A) and thus not seen at the 24 h time-point used in Figure 4.

### 3.3. Toxicity Is Caused by Amino Acid Accumulation, Not Proton Transport

The amino acid transporters studied here, as well as Gap1, are thought to be amino acid: proton symporters. One possible explanation for the toxicity of amino acid transport is that the corresponding influx of protons interferes with cellular homeostasis. This has been previously demonstrated in *E. coli* where excessive, uncontrolled transport of galactosides by the lactose:proton symporter LacY decreases the electrochemical proton gradient across the cell membrane and lowers the intracellular ATP concentration [81,82]. In the case of *S. cerevisiae* amino acid transport, it is possible that this effect would be amplified by the transport of excess amino acids into the vacuole, which is mediated by amino acid:proton antiporters would thus cause further movement of protons into the cytoplasm [83,84].

Risinger et al. [41] argued that proton influx was not the mechanism for Gap1-mediated amino acid toxicity by showing that growth was not inhibited by amino acid mixtures, a condition where the overall transport rate should remain high but individual amino acids would accumulate to lower levels. We did observe toxicity when cells were grown in a mixture of amino acids; in SC media, the growth of Can1 and Lyp1 strains was reduced 70% and 85%, respectively, while Dip5 did not have the growth advantage seen for the control and other overexpression strains (Figure 2). However, this likely reflects the narrow specificities of these transporters and their relatively low K*_m_*. The other amino acids are presumably not present in high enough concentrations to effectively compete for transport, and thus one or two substrates are still accumulated to high levels. Whole cell transport assays did not show any direct correlation between growth inhibition and initial transport rate (Figure 3), and this suggests that toxicity is not mediated by proton influx. Overexpression of Agp1 increased the transport of both Ala and Met 5- to 6-fold when each amino acid was supplied at 500 µM (Figure 3C,E), with the initial transport of Ala 3-fold higher than that of Met, yet at the same external concentration only Met inhibits growth. 50 µM of Lys is inhibitory to Lyp1 overexpression strains, yet the initial rate of transport is less than half that of the transport of Ala by Agp1 strains (Figure 3D,E). These transport assays also suggest differences in the intracellular concentration at which various amino acids become toxic. Assuming an internal volume of 70 fL per cell [85], we calculated that the amino acid pool of the Agp1 overexpression strain increased by 69 mM of Ala and 24 mM of Met in 24 min, and by 84 mM of Phe in 15 min. Under the same conditions, Met and Phe, but not Ala, cause growth inhibition of this strain.

To investigate the effect of increased proton flux more directly, we used a pH-sensitive GFP variant called ratiometric pHluorin [66,67] to monitor the cytoplasmic pH (pH_c_) in growing cells with or without the addition of amino acid to the media (Figure 5A). For these experiments, a truncation mutant of Lyp1 was used which is more stably maintained at the plasma membrane after lysine addition [62]. Overexpression of this variant also causes lysine sensitivity (data not shown). For control cells or Lyp1 cells without the addition of lysine, the pH_c_ was 7.0–7.2 and stayed fairly constant over three hours. In contrast, the pH_c_ of Lyp1 cells given 500 µM of lysine began to decrease after one hour, and continued to drop steadily over the course of the experiment with a total change of 0.4 pH units. Neither the magnitude of the pH change or the time scale on which it happens is consistent with the hypothesis of rapid proton influx as a causative mechanism. When yeast is exposed to glucose after a period of starvation, the cytoplasmic pH transiently drops to as low as pH 6 and cells are able to recover to normal levels within minutes [86,87]. Several groups have demonstrated a direct correlation between pH_c_ and growth rate in yeast, but *S. cerevisiae* is still able to grow reliably at pH_c_ between 6.5 and 7 [88,89].

We were able to obtain more information from the pHluorin experiments by analyzing the fluorescence signal at 425 nm excitation/508 nm emission. Under these conditions, pHluorin is insensitive to pH [67]. This means that we can monitor the amount of pHluorin independent of the pH changes. For the samples where pH_c_ remained constant, the bulk fluorescence fits well to an exponential curve (Figure 5B). We believe that this value is reporting the combined rates of protein production plus degradation in dividing cells, with the amount of pHluorin in individual cells staying stable over time. In contrast, the bulk fluorescence of Lyp1 cells after lysine addition begins to plateau at approximately the same time that the cytosolic pH begins to decrease. This suggests that the mechanisms leading to growth inhibition are occurring in the hour after lysine addition, and that the decrease in cytosolic pH is a consequence, rather than a cause.

One hypothesis that would fit our data is that lysine (and also cysteine) toxicity is due to interference with ubiquitination pathways. Protein modification by the attachment of ubiquitin (Ub) is a regulatory mechanism involved in a wide range of essential processes in eukaryotic cells [90]. A key step in these pathways is the transfer of Ub from a Ub-conjugating (E2) enzyme to a target protein. Some human E2~Ubs are able to react with free lysine and cysteine molecules in such a way that Ub is irreversibly transferred to the amino acid [91,92]. Intrinsic reactivity with free lysine has also been observed for the *S. cerevisiae* E2 enzymes Ubc4 and Pex4 (Chris Williams, personal communication). This reaction occurs on the minute timescale, in the cytoplasmic pH range and at concentrations as low as 50 mM. Extrapolation of our transport assay data suggests that our overexpression strains import this amount of lysine within 30–60 min, which is in agreement with previous work [62]. Overlapping activities mean that individual E2 enzymes are not essential in *S. cerevisiae* but multiple knockouts, for example *Δubc4Δubc5*, are lethal [93].

## 4. Conclusions

Our study expands on the work of Risinger et al. [41] who showed that all 20 proteinogenic amino acids, as well as the non-proteinogenic amino acids citrulline and ornithine, can be growth inhibitory to *S. cerevisiae*. We have shown that this effect can be mediated by various amino acid transporters and is not specific to Gap1. We have also demonstrated that amino acid-mediated growth inhibition is not dependent on the initial rate of transport, or triggered by the rapid influx of protons, but is instead caused by the longer-term accumulation of single amino acids. For at least some amino acids, *S. cerevisiae* has mechanisms in place to prevent their over-accumulation by substrate-dependent removal of the corresponding transporters from the plasma membrane and/or partitioning in specific membrane domains (as shown for example for Can1 and Lyp1) [94,95,96].

Amino acid sensitivity is an important phenomenon that should be considered in the design and analysis of studies of amino acid and peptide transport. It is also a useful tool for assessing the in vivo activity of transporters as it reports on their levels at the plasma membrane and their transport kinetics for specific substrates. We have used it to develop growth-based screens to confirm the activity of overexpressed wild-type and mutant transporters, including when expressed in *Pichia pastoris* [62]. Our results from screening eight different amino acid transporters did vary from what we predicted based on the current literature suggesting (i) a much broader specificity than previously thought, and (ii) transporter/substrate-specific variations that may reflect differences in substrate-induced downregulation.

## Figures and Tables

**Figure 1 microorganisms-09-00007-f001:**
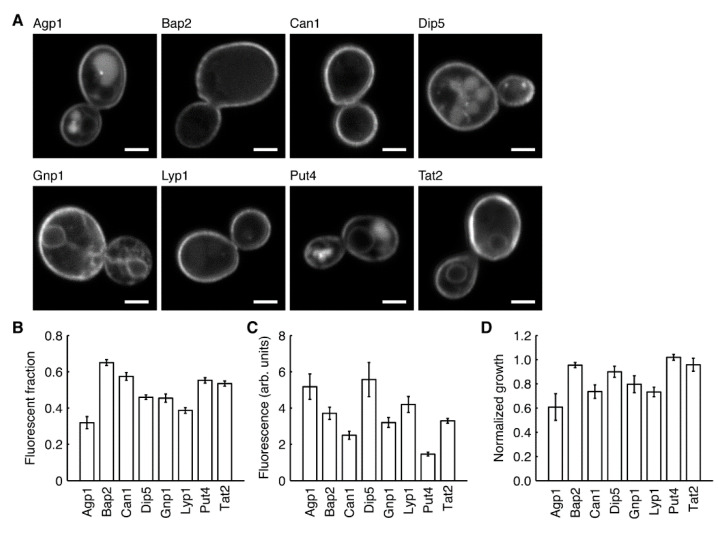
Overexpression of amino acid permeases in BY4709. All proteins contained a C-terminal YPet tag and were expressed from a multicopy plasmid under control of the constitutive ADH1 promoter. (**A**) Fluorescence confocal microscopy images showing that the transporters localize mainly to the cell periphery. Scale bars are 2 µm. (**B**,**C**) Data from flow cytometry analysis showing the estimated fraction of fluorescent cells in each culture (**B**), and the median fluorescence signal of cells in this sub-population (**C**). (**D**) Growth of overexpression strains in minimal media. Normalized growth refers to cell density after 24 h, normalized to BY4709 carrying the empty plasmid pRSII426. Values shown are the mean ± standard deviation (*n* = 5 for (**B**,**C**), *n* = 7 for (**D**)).

**Figure 2 microorganisms-09-00007-f002:**
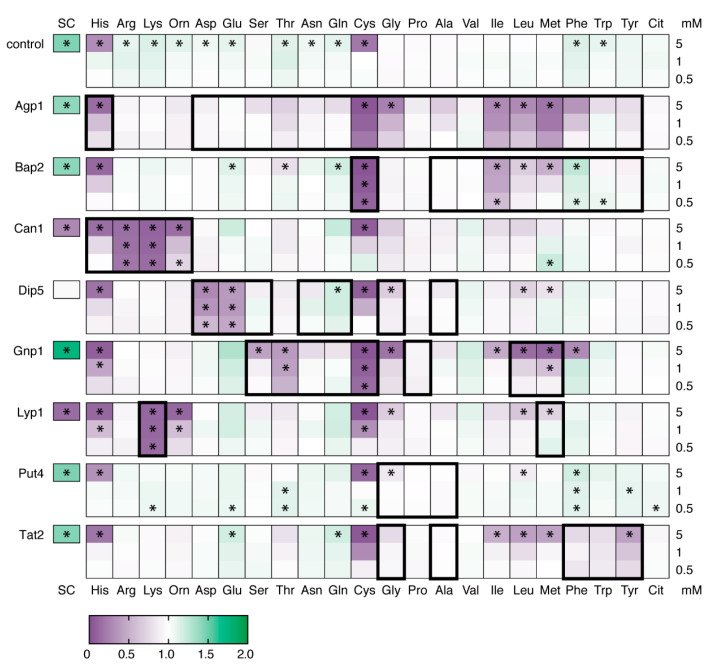
The effect of amino acids on the growth of strains overexpressing amino acid transporters. BY4709 carrying the empty plasmid pRSII426 was used as a control. All proteins were expressed from the constitutive ADH1 promoter with a C-terminal YPet tag. Heat maps show cell density (OD_600_) after 24 h, normalized such that the same strain in YNBD minimal media without amino acids = 1. Raw values are given in Appendix A. The proteinogenic amino acids are represented by their single letter code (see Table 4). Orn = ornithine, Cit = citrulline, SC = synthetic complete media minus uracil. Amino acids were added at final concentrations of 0.5, 1, or 5 mM. Thick black boxes indicate reported substrates for each transporter (for references, see Table 1). Asterisks (*) indicate *p* < 0.05 when compared to growth in YNBD (*t*-test).

**Figure 3 microorganisms-09-00007-f003:**
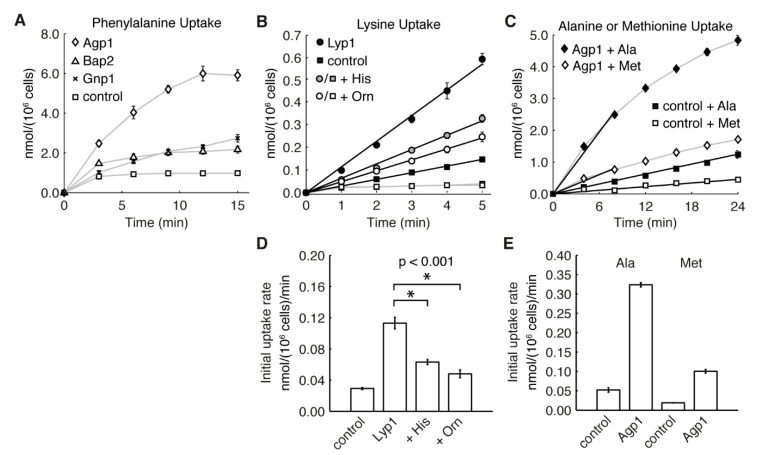
Amino acid uptake by *S. cerevisiae* cells. Transporters were expressed from the constitutive ADH1 promoter with a C-terminal YPet tag. BY4709 carrying the empty plasmid pRSII426 was used as a vector control strain. (**A**) Uptake of phenylalanine (2.5 mM). (**B**) Uptake of lysine (50 µM) in the absence (black symbols) or presence of histidine (100 mM, grey symbols) or ornithine (100 mM, white symbols); circles, Lyp1 overexpression; squares, vector control strain. (**C**) Uptake of either alanine (500 µM, black symbols) or methionine (500 µM, white symbols); diamonds, Agp1 overexpression; squares, vector control strain. The straight black lines in (**B**,**C**) represent the calculated initial uptake rates, which are shown in (**D**,**E**), respectively. Values are the mean of biological triplicates. Error bars represent standard deviation and, in some cases, are obscured by the data point. Rates in (**D**) were compared using a *t*-test, and asterisks (*) represent *p* < 0.001.

**Figure 4 microorganisms-09-00007-f004:**
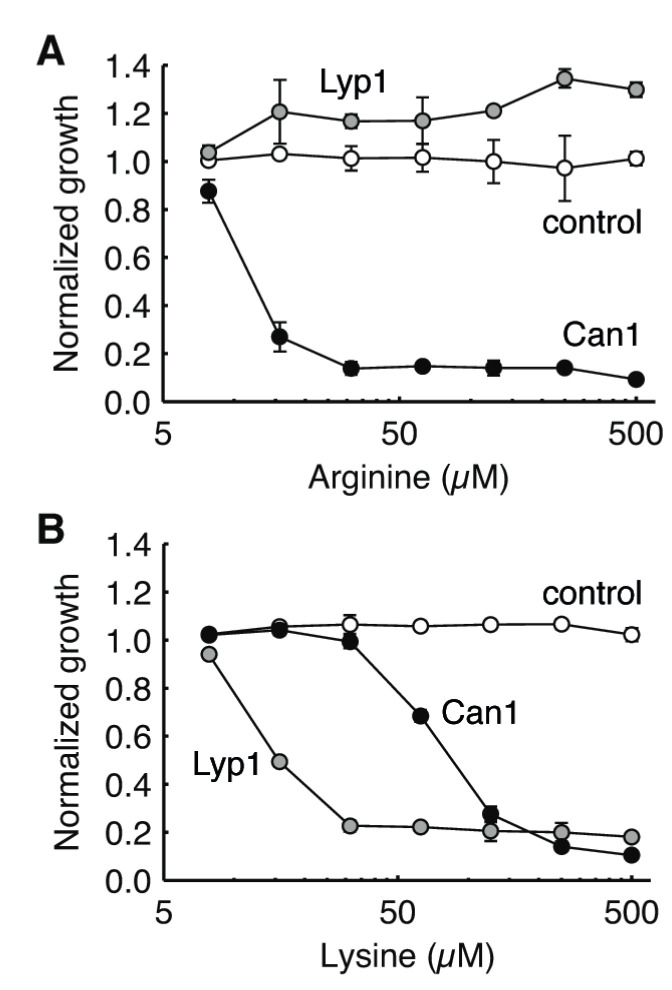
The effect of external arginine (**A**) and lysine (**B**) on the growth of strains overexpressing Can1 or Lyp1. BY4709 carrying the empty plasmid pRSII426 was used as a control. Can1 and Lyp1 were expressed from the constitutive ADH1 promoter with a C-terminal YPet tag. Normalized growth refers to cell density after 24 h, normalized to minimal media without amino acids. Values are the mean of biological triplicates. Error bars represent standard deviation and, in some cases, are obscured by the data point.

**Figure 5 microorganisms-09-00007-f005:**
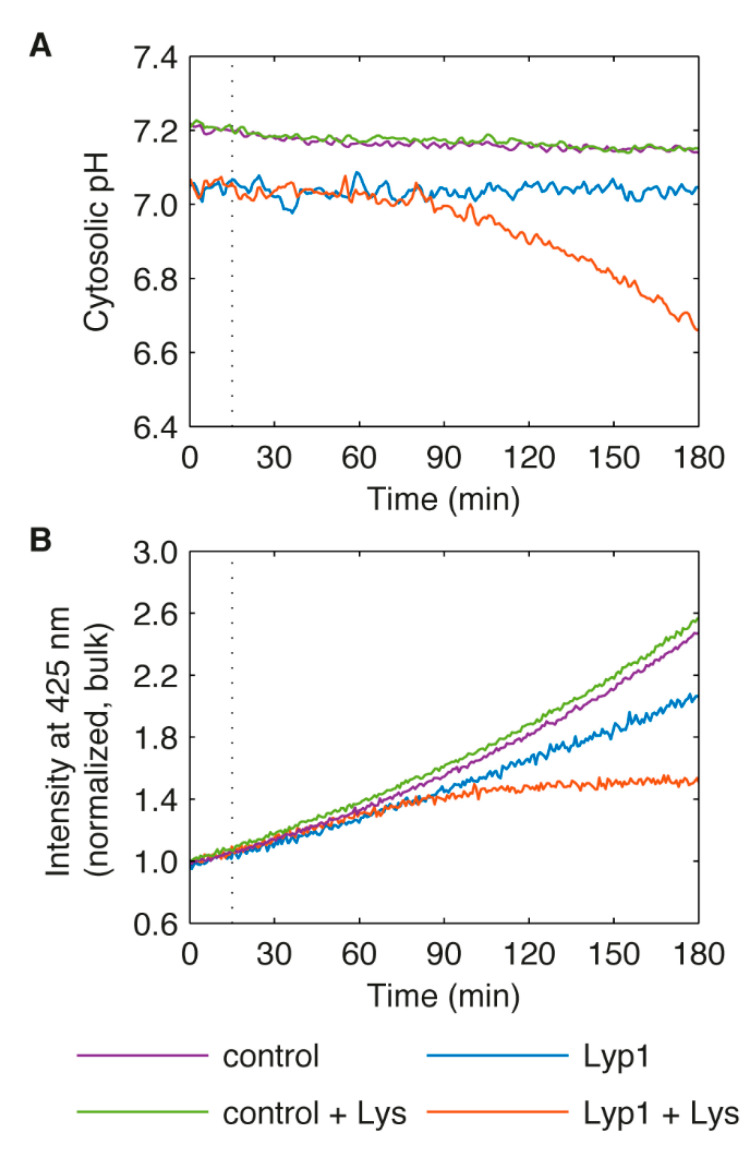
The effect of external lysine on the cytosolic pH of cells overexpressing Lyp1. BY4741 expressing the pH-sensitive ratiometric pHluorin and carrying the empty plasmid pRSII425 was used as a control. The Lyp1 strain constitutively expresses both pHluorin and a truncated, non-fluorescently tagged Lyp1. (**A**) Cytosolic pH. (**B**) Normalized bulk intensity at 425 nm excitation/508 nm emission, at which pHluorin is largely insensitive to pH. Values shown are the mean of two independent experiments (individual results are shown in Appendix A). Dotted lines indicate the time at which either distilled water or lysine (final concentration 500 µM) was added.

**Table 1 microorganisms-09-00007-t001:** Transporters used in this study and their reported substrates. Values in brackets indicate published Michaelis constants (*K_m_*).

Name	Transport Substrates	Reference(s) *
Agp1	His, Asp, Glu, Ser, Thr, Asn (0.29 mM), Gln (0.79 mM), Cys, Gly, Pro, Ala, Val, Ile (0.6 mM), Leu (0.16 mM), Met, Phe (0.6 mM), Tyr, Trp	[42,43,44,45,46]
Bap2	Cys, Ala, Val, Ile, Leu (37 µM), Met, Phe, Tyr, Trp	[46,47,48,49]
Can1	His, Arg (10–20 µM), Lys (150–250 µM), Orn	[50,51]
Dip5	Glu (48 µM), Asp (56 µM), Ser, Asn, Gln, Gly, Ala	[52]
Gnp1	Ser, Thr, Asn, Gln (0.59 mM), Cys, Pro, Leu, Met	[44,45,53]
Lyp1	Lys (10–25 µM), Met	[50,51,54,55]
Put4	Gly, Pro, Ala	[45,56,57,58]
Tat2	Gly, Ala, Phe, Tyr, Trp, Cys	[44,59]

* All proteins in this table were studied in Regenberg et al. [60].

**Table 2 microorganisms-09-00007-t002:** Plasmids used in this study.

Name	Description	Source
pRSII425	*LEU2*, 2μ (multicopy)	[65]
pRSII426	*URA3*, 2μ (multicopy)	[65]
pFB022	pRS426 (*URA3*, 2μ) derivative with Lyp1YPet under ADH1 promoter	[62]
pFB023	pFB022 derivative containing Lyp1_(62-590)_YPet	[62]
pSR053	pFB023 derivative containing Lyp1_(62-590)_	This study
pSR045	pFB022 derivative containing Agp1YPet	This study
pSR046	pFB022 derivative containing Bap2YPet	This study
pSR047	pFB022 derivative containing Dip5YPet	This study
pSR048	pFB022 derivative containing Gnp1YPet	This study
pSR049	pFB022 derivative containing Put4YPet	This study
pSR050	pFB022 derivative containing Tat2YPet	This study
pSR051	pFB022 derivative containing Can1YPet	This study
pSR057	pRSII425 derivative containing Lyp1_(62-590)_ under ADH1 promoter	This study
pYES2-*P_ACT1_*-pHluorin	pYES2 (*URA3*, 2μ) derivative with pHluorin under ACT1 promoter	[66]

**Table 3 microorganisms-09-00007-t003:** Oligonucleotide primers used in PCR amplification.

Name	Sequence (5′ to 3′)	Description ^1^
5273	GGAGGGGAAAATTTATATTTTCAAGGTTC	pFB022 (F)
5954	CATTTTGGGATCCACTAGTTCTAG	pFB022 (R)
5480	TCTAGAACTAGTGGATCCCAAAATGTCGTCGTCGAAGTCTC	*AGP1* (F)
5481	TCTAGAACTAGTGGATCCCAAAATGCTATCTTCAGAAGATTTTGGATC	*BAP2* (F)
5561	TCTAGAACTAGTGGATCCCAAAATGGGAACAAATTCAAAAGAAG	*CAN1* (F)
5482	TCTAGAACTAGTGGATCCCAAAATGAAGATGCCTCTAAAGAAGATG	*DIP5* (F)
5483	TCTAGAACTAGTGGATCCCAAAATGACGCTTGGTAATAGACGC	*GNP1* (F)
5484	TCTAGAACTAGTGGATCCCAAAATGGTAAATATACTGCCCTTCC	*PUT4* (F)
5485	TCTAGAACTAGTGGATCCCAAAATGACCGAAGACTTTATTTCTTCTG	*TAT2* (F)
5486	ACCTTGAAAATATAAATTTTCCCCTCCACACCAGAAGGCAACGAC	*AGP1* (R)
5487	ACCTTGAAAATATAAATTTTCCCCTCCACACCAGAAATGATAAGCTTTTCTC	*BAP2* (R)
5562	ACCTTGAAAATATAAATTTTCCCCTCCTGCTACAACATTCCAAAATTTG	*CAN1* (R)
5488	ACCTTGAAAATATAAATTTTCCCCTCCGAAGATATTACCCAAAAATTTTTCATAG	*DIP5* (R)
5489	ACCTTGAAAATATAAATTTTCCCCTCCACACCAGAAATCAAGAACTCTTTTC	*GNP1* (R)
5490	ACCTTGAAAATATAAATTTTCCCCTCCCAACAAGGCGTCCAAGAAC	*PUT4* (R)
5491	ACCTTGAAAATATAAATTTTCCCCTCCACACCAGAAATGGAACTGTCTC	*TAT2* (R)
4258	ACCACCACCAUCATCATCATCATTAACTGCAGGAATTC	pFB023-A (F)
3631	AGCACTACCCUTTAGCTGTTCTATATGCTGCC	pFB023-A (R)
3630	AGGGTAGTGCUGAAGGAAGCATACGATACCC	pFB023-B (F)
5159	ATTTTGGGAUCCACTAGTTCTAGAGCGGCCAGCTTGGAGTTGATTG	pFB023-B (R)
5087	ATCCCAAAAUGCATGGGTCATTGCAAGGTGG	*LYP1_(62-590)_* (F)
5089	ATGGTGGTGGUGTCCCCCTCCTTCGATTTCTCTTCTGTCGGAATC	*LYP1_(62-590)_* (R)

^1^ F = forward primer, R = reverse primer.

**Table 4 microorganisms-09-00007-t004:** Concentration (mM) of components in synthetic complete (SC) mixture, when used at 1x.

**Amino Acids (3-letter, 1-letter code)**	
Alanine (Ala, A)	0.853
Arginine (Arg, R)	0.361
Asparagine (Asn, N)	0.575
Aspartic acid (Asp, D)	0.571
Cysteine (Cys, C)	0.627
Glutamine (Gln, Q)	0.520
Glutamate (Glu, E)	0.517
Glycine (Gly, G)	1.012
Histidine (His, H)	0.490
Isoleucine (Ile, I)	0.579
Leucine (Leu, L)	2.897
Lysine (Lys, K)	0.520
Methionine (Met, M)	0.509
Phenylalanine (Phe, F)	0.460
Proline (Pro, P)	0.460
Serine (Ser, S)	0.723
Threonine (Thr, T)	0.638
Tryptophan (Trp, W)	0.372
Tyrosine (Tyr, Y)	0.419
Valine (Val, V)	0.649
**Other**	
Adenine	0.133
*myo-*Inositol	0.422
4-Aminobenzoic acid	0.058

## Data Availability

Please refer to suggested Data Availability Statements in section “MDPI Research Data Policies” at https://www.mdpi.com/ethics.

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
