# Peer review of "Growth Inhibition by Amino Acids in Saccharomyces cerevisiae"

_microorganisms, 2020, doi:10.3390/microorganisms9010007_

Round 1

Reviewer 1 Report

The manuscript by Ruiz and colleagues reports an interesting investigation on the Saccharomyces cerevisiae growth inhibition induced by proteinogenic aminoacids. The experimental design and the investigation are scientifically sound, and the manuscript is clearly written.

The findings of this study will be fundamental for future investigations but also to interpreting more properly previous studies dissecting functions related to aminoacid metabolism and S. cerevisiae physiology.

I have just I minor comment on the evolutionary implications of the results of this study. Despite the authors having commented the results in the optic of experimental settings, I believe that contextualizing the findings in the evolution of the budding yeast would be interesting. Possibly beyond the scope of this study, it would be interesting to observe, if not already done, the level of the endogenous expression of the investigated transporters in the presence of the tested aminoacid concentrations. Would the expression of the transporter regulated in order to avoid the uptake of aminoacid to a toxic intracellular concentration? This would be evolutionarily reasonable… Despite capable of synthesizing aminoacids, wild S. cerevisiae strains would need N sources to synthesize them, and synthesizing aminoacids requires energy too. So, being capable of using environmental aminoacids would represent an advantage- and so it would be expressing high levels of transporters, unless the intracellular concentration of the aminoacid reaches the toxic level. Similarly, the intake of glucose in S. cerevisiae is finely regulated and mediated by transporters showing different affinities. Would it be the same with aminoacids? Are there other systems to prevent accumulation of these above toxic levels?

In the same optic, are the tested aminoacids ever found at the higher tested concentrations in nature? Or are these concentrations an artifact necessary to test in vitro the hypothesis?

L206: please correct “was reasonably stable…” to “were reasonably stable…”

Author Response

Reviewer 1

The manuscript by Ruiz and colleagues reports an interesting investigation on
the Saccharomyces cerevisiae growth inhibition induced by proteinogenic aminoacids. The experimental design and the investigation are scientifically sound, and the manuscript is clearly written. The findings of this study will be fundamental for future investigations but also to interpreting more properly previous studies dissecting functions related to aminoacid metabolism and S. cerevisiae physiology.

Reply: We would like to thank Reviewer 1 for his/her comments and suggestions.

I have just I minor comment on the evolutionary implications of the results of this study. Despite the authors having commented the results in the optic of experimental settings, I believe that contextualizing the findings in the evolution of the budding yeast would be interesting. Possibly beyond the scope of this study, it would be interesting to observe, if not already done, the level of the endogenous expression of the investigated transporters in the presence of the tested
aminoacid concentrations. Would the expression of the transporter regulated in order to avoid the uptake of aminoacid to a toxic intracellular concentration? This would be evolutionarily reasonable… Despite capable of synthesizing aminoacids, wild S. cerevisiae strains would need N sources to synthesize them, and synthesizing aminoacids requires energy too. So, being capable of using environmental aminoacids would represent an advantage- and so it would be
expressing high levels of transporters, unless the intracellular concentration of the aminoacid reaches the toxic level. Similarly, the intake of glucose in S. cerevisiae is finely regulated and mediated by transporters showing different affinities. Would it be the same with aminoacids? Are there other systems to prevent accumulation of these above toxic levels? In the same optic, are the tested aminoacids ever found at the higher tested concentrations in nature? Or are these concentrations an artifact necessary to test in vitro the hypothesis? L206: please correct “was reasonably stable…” to “were reasonably stable…”

Reply: The reviewer presents some very intriguing and interesting ideas. We would expect that the expression of amino acid transporters is tuned to the availability of amino acids, as we and others have shown for Lyp1 and Can1. Indeed, there are parallels with how glucose intake is regulated in yeast. We now emphasize this possibility and included three new references in the Conclusions section of the revised manuscript (page 15, lines 397-400). We feel that it is beyond
the scope of the present manuscript to test the hypothesis for all “toxic” amino acids as it would require fluorescent tagging of chromosomally-expressed genes or raising antibodies against purified transporters, which would require more than one year of work.

Reviewer 2 Report

This article addresses and important facet in cell biology, amino acid homeostasis. It has been known for a long time that dysregulated uptake of some amino acids can be toxic. This manuscript expands on this topic by exploring the toxicity of different amino acids when combined with overexpression of transporters in yeast. This manuscript makes a nice addition to the literature by 1) showing overexpression of many transporters can be toxic in the presence of external amino acids, 2) showing that the amino acid specificity of some transporters is more broad that currently appreciated at amino acid concentrations commonly used in the lab environment, and 3) providing additional support for the models that amino acid levels themselves are toxic rather than low pH of the cytosol.

This work is significant for many fields including the membrane trafficking and amino acid signaling field which control amino acid transporters and thus may impinge on amino acid toxicity in certain situations.

The conclusions are well supported by the data and appropriate controls have been done.

The authors have accurately described some of the shortcomings of the experimental design which uses plasmids that can be lost or lead to cell-to-cell variation in the overexpression. Although a different experimental design might have been ideal, given the controls provided and consideration of this shortcoming by the authors, I think the experimental design acceptable.

Introduction is extremely well written and referenced.

Materials and methods are clear. I commend the authors on their extensive documentation of resources including oligos, plasmids, strains used, and exact amounts of amino acids in the media. I also commend them on their clear description of the pH calibration and required controls.

I have some minor suggestions to improve the manuscript if the authors choose to do so:

1) Although the authors nicely measure pH changes, and indicate that the timeframe of changes might not explain the toxicity, I wonder if they have considered other changes in cytosolic chemistry that could explain the difference. Is it possible that hyperactive transporters are causing a depletion of ATP, or by disrupting the pH gradient altering the ability of cells to internalize something other than amino acids? The authors might consider discussing these possibilities

2) I suggest the authors provide a reference for the Ypet tag in the first mention in the results and discussion, for readers who might want ready access to that without looking through the materials and methods.

3) I struggled to understand the labels on figure 3b looking at lyp1 overexpression in the presence of exogenous his/orn. Perhaps a more informative legend would be useful. I had a similar issue with figure 3c.

4) I found the results and discussion somewhat difficult to follow. Each paragraph contained a lot of information, and sometimes more than one concept. The authors might consider breaking up their paragraphs to contain only one concept at a time and giving themselves the space to more fully explain their logic and interpretations.

Author Response

Reviewer 2

This article addresses and important facet in cell biology, amino acid homeostasis. It has been known for a long time that dysregulated uptake of some amino acids can be toxic. This manuscript expands on this topic by exploring the toxicity of different amino acids when combined with overexpression of transporters in yeast. This manuscript makes a nice addition to the literature by 1) showing overexpression of many transporters can be toxic in the presence of external amino acids, 2) showing that the amino acid specificity of some transporters is more broad that currently appreciated at amino acid concentrations commonly used in the lab environment, and 3) providing additional support for the models that amino acid levels themselves are toxic rather than low pH of the cytosol.

This work is significant for many fields including the membrane trafficking and amino acid signaling field which control amino acid transporters and thus may impinge on amino acid toxicity in certain situations.

The conclusions are well supported by the data and appropriate controls have been done.

The authors have accurately described some of the shortcomings of the experimental design which uses plasmids that can be lost or lead to cell-to-cell variation in the overexpression. Although a different experimental design might have been ideal, given the controls provided and consideration of this shortcoming by the authors, I think the experimental design acceptable.

Introduction is extremely well written and referenced.

Materials and methods are clear. I commend the authors on their extensive documentation of resources including oligos, plasmids, strains used, and exact amounts of amino acids in the
media. I also commend them on their clear description of the pH calibration and required controls.

Reply: We also would like to thank Reviewer 2 for his/her comments and suggestions.

I have some minor suggestions to improve the manuscript if the authors choose to do so:

1) Although the authors nicely measure pH changes, and indicate that the timeframe of changes might not explain the toxicity, I wonder if they have considered other changes in cytosolic chemistry that could explain the difference. Is it possible that hyperactive transporters are causing a depletion of ATP, or by disrupting the pH gradient altering the ability of cells to internalize something other than amino acids? The authors might consider discussing these possibilities

Reply: We have considered alternative possibilities. The maintenance of the cytosolic pH is a direct measure of the pH gradient and connected to the ATP levels of the cell. As indicated in the paper (pages 13-14) we believe that the mechanisms leading to growth inhibition are occurring in the hour after lysine addition (and are not due to depletion of ATP or disruption of the pH gradient), and that the decrease in cytosolic pH is a consequence, rather than a cause.

2) I suggest the authors provide a reference for the Ypet tag in the first mention in the results and discussion, for readers who might want ready access to that without looking through the materials and methods.

Reply: We included a link to the website where information on YPet can be found:
(https://www.fpbase.org/protein/ypet/)

3) I struggled to understand the labels on figure 3b looking at lyp1 overexpression in the presence of exogenous his/orn. Perhaps a more informative legend would be useful. I had a similar issue with figure 3c.

Reply: We thank the reviewer for his/her comments. We have modified the figure expanded the legend to make the presentation more clear.

4) I found the results and discussion somewhat difficult to follow. Each paragraph contained a lot of information, and sometimes more than one concept. The authors might consider breaking up their paragraphs to contain only one concept at a time and giving themselves the space to more fully explain their logic and interpretations.

Reply: We have subdivided the Results and Discussion in three sections (see below), which are the key messages. We feel that a further subdivision would distract from the message we would like to convey in these paragraphs.
3.1. All standard proteinogenic amino acids, as well as ornithine and citrulline, inhibit growth of S. cerevisiae
3.2. Amino acid sensitivity reports on transporter activity and substrate specificity
3.3. Toxicity is caused by amino acid accumulation, not proton transport